# Inversion of HF Radar Doppler Spectra Using a Neural Network

**Rachael L. Hardman *** and **Lucy R. Wyatt**

Schools of Mathematics and Statistics, University of Sheffield, Sheffield S3 7RH, UK
* Correspondence: rlhardman1@sheffield.ac.uk

**Abstract:** For a number of decades, coastal HF radar has been used to remotely measure ocean surface parameters, including waves, at distances exceeding 100 km. The information, which has value in many ocean engineering applications, is obtained using the HF radar cross-section, which relates the directional ocean spectrum to the received radar signal, through a nonlinear integral equation. The equation is impossible to solve analytically, for the ocean spectrum, and a number of numerical methods are currently used. In this study, a neural network is trained to infer the directional ocean spectrum from HF radar Doppler spectra. The neural network is trained and tested on simulated radar data and then validated with data collected off the coast of Cornwall, where there are two HF radars and a wave buoy to provide the sea-truth. Key ocean parameters are derived from the estimated directional spectra and then compared with the values measured by both the wave buoy and an existing inversion method. The results are encouraging; for example, the RMSE of the obtained mean wave direction decreases from 20.6° to 15.7°. The positive results show that neural networks may be a viable solution in certain situations, where existing methods struggle.

**Keywords:** HF radar; artificial neural network; remote sensing; inversion; radar cross-section; monostatic radar; ocean wave directional spectrum; TensorFlow

---

## 1. Introduction

By employing the perturbation method of Rice [1], Barrick [2,3] derived the expected monostatic radar cross-section of the ocean surface $\sigma(\omega)$, given an ocean spectrum $S(\vec{k})$ and radar wavenumber $k_0$. The expression is written as:

$$\sigma(\omega) = \sigma^{(1)}(\omega) + \sigma^{(2)}(\omega), \tag{1}$$

where $\sigma^{(1)}(\omega)$ signifies the first order scatter and $\sigma^{(2)}(\omega)$ the second order. Explicitly, the first order contribution is:

$$\sigma^{(1)}(\omega) = 2^6 \pi k_0^4 \sum_{m=\pm 1} S(m\vec{k}_B)\delta(\omega - m\omega_B), \tag{2}$$

and is due to Bragg scattering from ocean waves with half the radio wavelength, or equivalently wavenumber $k_B = 2k_0$, travelling directly towards or away from the radar. When there is no ocean current, the first order contribution appears as two dominant peaks, as described by the delta function $\delta()$, in the Doppler spectrum at the frequencies $\pm\omega_B$, where:

$$\omega_B = \sqrt{2gk_0 \tanh(2k_0 d)} \tag{3}$$

is known as the *Bragg frequency*, for ocean depth $d$ and gravity $g$. If a current *is* present, the whole spectrum is subject to a Doppler shift, proportional to the current speed and direction. HF radar has been used as a remote measuring device of ocean currents for decades; see the works of Paduan and Washburn [4] and Abascal et al. [5] for further details.

The second order term,

$$\sigma^{(2)}(\omega) = 2^6 \pi k_0^4 \sum_{m,m'=\pm} \iint_{-\infty}^{\infty} |\Gamma_T|^2 S(m\vec{k}_1) S(m'\vec{k}_2) \delta(\omega - m\omega_1 - m'\omega_2) \, dp \, dq, \tag{4}$$

describes the second order combinations of the electromagnetic and hydrodynamic waves. In $\sigma^{(2)}(\omega)$, $\vec{k}_1$ and $\vec{k}_2$ are the two contributing wave vectors (with respective angular frequencies $\omega_1$ and $\omega_2$), which sum to give $\vec{k}_B$. $|\Gamma_T|^2$ is the *coupling coefficient*, which is a function of $\vec{k}_1$ and $\vec{k}_2$ and contains the mathematics of the nonlinear combinations. More detail on the coupling coefficient can be found in the work of Lipa and Barrick [6], amongst many others. The second order contribution surrounds the first order peaks and will henceforth be referred to as the *sidebands*.

Numerical methods, such as those described by Holden and Wyatt [7], can be used to calculate the value of $\sigma(\omega)$ for a measured, or modelled, $S(\vec{k})$. For this work, a Python code was written to simulate the radar data, using a vectorised version of the code presented by Lipa and Barrick [8]. An example of a simulated radar Doppler spectrum, along with a measured radar Doppler spectrum, is shown in Figure 1. To validate the radar cross-section expression, a value for $S(\vec{k})$, measured by the wave buoy described in Section 2.1.2, has been used as input to the simulation. The resulting Doppler spectrum has then been compared to a radar Doppler spectrum (measured by the radar described in Section 2.1.1), measured at the same time and location as the wave buoy. The two spectra are satisfactorily similar, with the most noticeable differences at frequencies near zero-Doppler. This is because values near zero-Doppler are not calculated in the simulation code, to save time, as they are not used in any inversion methods.

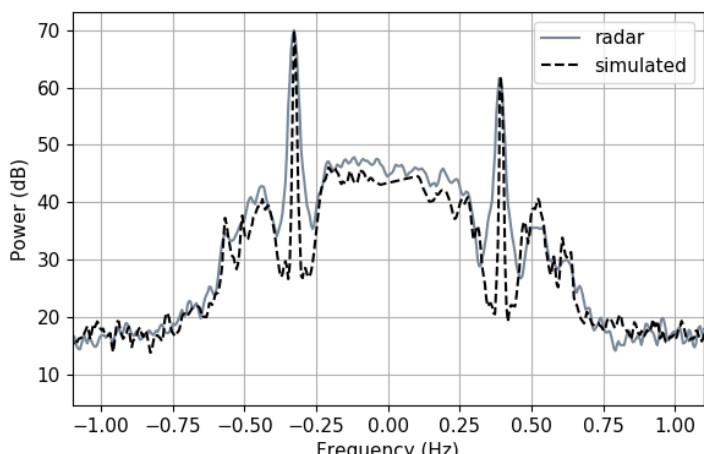

**Figure 1.** Comparison of measured and simulated monostatic Doppler spectra on 1 November 2012 20:05. The figure shows a comparison of a radar-measured Doppler spectrum from the Wave Hub HF dataset at the location of the wave buoy and a Doppler spectrum simulated using Equations (2) and (4) with $S(\vec{k})$ from the wave buoy input.

The inversion of Equation (4), for $S(\vec{k})$, allows us to measure the wave spectrum remotely. This has important applications in numerous coastal engineering topics, including the testing of and assimilation into operational wave models, supporting vessel navigation and monitoring both renewable energy sources and climate change. The works of Lipa [9,10], Wyatt [11], Green and Wyatt [12], Howell and Walsh [13] and Hisaki [14] are notable in the history of inverting Equation (4) numerically. Empirical methods such as that of Gurgel et al. [15], who empirically fit wave height

to the second order sidebands using a wave buoy, are also possible, if one wants to avoid inverting the equation.

In this paper, a new inversion method is proposed, which uses a neural network to invert the Doppler spectrum (see Hardman [16] for full details). The neural network was trained on simulated Doppler spectra, generated using the radar cross-section expression given in Equations (2) and (4), using directional spectra measured from a local wave buoy and modelled using parameters from a WAVEWATCH III [17] model as the input. The performance of the method was measured against the existing method of Wyatt [11] and Green and Wyatt [12], henceforth referred to as the *Seaview method*, which has been validated in a number of works (e.g., [18–20]). To compare the two methods, both have been used to invert the same set of measured HF radar data, where a wave buoy is used to provide the sea-truth.

The majority of the existing inversion methods rely on the radar cross-section expression given in Equations (2) and (4). Thus, the methods produce good results when the data are accurately modelled by this expression. However, as was shown by Wyatt et al. [21], this was not always the case. For phased array receivers, using a beamforming method, large sidelobes can occur. Large sidelobes, coupled with a strong ocean current, or (as shown by Grosdidier et al. [22] for bistatic radar setups) low radar resolution can mean that the data are not modelled by the derived radar cross-section and, consequently, is not appropriate for inversion using the existing methods. A neural network learns by example, so if the effects of the radar data are included in the data that it learns from, then it should theoretically be able to model the relationship between the radar Doppler spectra and the corresponding ocean spectrum $S(\vec{k})$. Motivated by the goal of inverting data that cannot be modelled by the radar cross-section expression, this work is an experiment to see if a neural network can successfully invert high-quality HF Doppler spectra, i.e., when the data *are* representative of the radar cross-section equations. Therefore, no additional averaging or sidelobes were included in the simulations at this stage.

Although this is the first time that a neural network has been used to invert Doppler spectra to obtain directional ocean spectra, they have previously been used to measure wind speeds and water levels from HF radar data. Mathew and Deo [23] trained a neural network with HF-measured values of significant wave height, wave period, wave direction and wind direction input, to predict wind speed as measured by a wave buoy. Zeng et al. [24] did the same, but removed wave direction from the input data. Similarly, Shen et al. [25] trained a neural network to predict wind speed using the measurements of a local wave buoy and HF-measured first order Bragg peaks. To measure sea level, Wahle and Stanev [26] trained a neural network on tidal gauge data and HF radar current measurements. Seemingly all of the previous neural network applications in the field have used other measuring devices, such as wave buoys or tidal gauges, alongside measured radar data, to train their algorithms. In this work, by using simulations, the need for another device has been eradicated, and the HF radar system has no dependence on other devices.

We begin by describing the experimental setup and the necessary datasets in Section 2.1. Outlines of the new and existing methods for wave inversion, tested in this work, are described in Section 2.2, before the results of both methods are presented in Section 3. A discussion including the limitations of the method and scope for future work is given in Section 4, and concluding comments follow in Section 5.

## 2. Materials and Methods

### 2.1. Data

#### 2.1.1. HF Radar

In the experiment, HF radar data, obtained by two monostatic WERAHF radar systems [27] on the north coast of Cornwall, SW England, were used to validate the inversion methods. The radars, which are operated by Plymouth University, were set up to aid the experiments at a test site for offshore

renewable energy, known as Wave Hub. A full description of the site and results of the experiment were given by Lopez and Conley [28]. At the experiment site, shown in Figure 2, one radar is situated at Pendeen (50.16° N, 5.67° W), and a second radar is situated, approximately 40 km away, at Perranporth (50.33° N, 5.18° W). Both radars operate at approximately 12.3 MHz. Radar data from November 2012 were used where hourly measurements were available. Each transmitter consisted of 4 elements arranged in a square, and each phased array receiver consisted of 16 antennas. The received signals were digitally beamformed onto a rectangular grid with a resolution of 1 km per cell. This dataset is referred to in this work as the *Wave Hub HF radar data*.

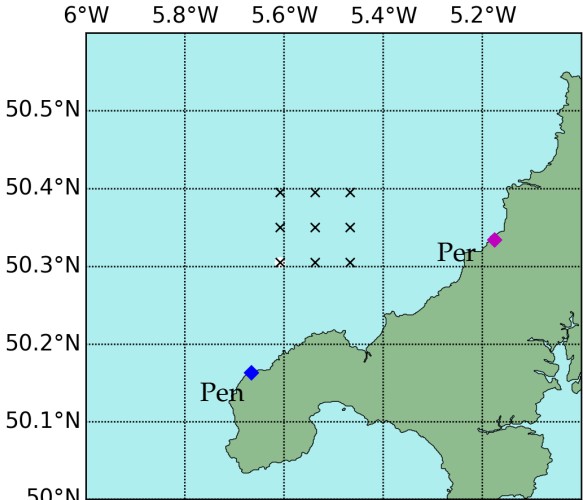

**Figure 2.** The Pendeen radar is shown by the blue diamond and the Perranporth radar by the magenta diamond. The wave buoy is shown by the white circle, and the location of the radar data used to validate the inversion method is shown by the black crosses.

### 2.1.2. Wave Buoy

A directional wave buoy situated in the radar coverage area (50.31° N, 5.61° W), at a location where the ocean depth is 53 m, was used to provide the sea-truth. Approximately 20 km from the Pendeen radar and 30 km from the Perranporth radar, measurements from the "Seawatch Mini II directional buoy" were in 30-min intervals, and the directional spectra were calculated using a maximum entropy method (see Lopez et al. [29] for details). Two different datasets from the wave buoy were used in this study; one for simulating radar Doppler spectra and one for validating the method. The two datasets overlapped, every hour, in November 2012, so to avoid bias in the neural network, the overlapping data were removed from the simulation dataset. The specifications of both wave buoy datasets are given in Table 1.

**Table 1.** Specifications of the wave buoy datasets.

|  | **Simulation** | **Validation** |
| --- | --- | --- |
| Dates | 1 September 2012–20 December 2012 | 1–30 November 2012 |
| Time resolution | 30 min (with data overlapping the validation set removed) | 60 min |
| Frequency resolution | 0.0078 Hz in [0–1 Hz] (129 measurements) | 0.0078 Hz in [0–1 Hz] |
| Directional resolution | 12° in [0–360] (30 measurements) | 4° in [0–2$\pi$] (90 measurements) |
| Data points | 4830 | 648 |

### 2.1.3. WAVEWATCH III

WAVEWATCH III (WW3) data (see Tolman et al. [17]), from the IOWAGA database [30], were used in this experiment, alongside the wave buoy data, to simulate Doppler spectra. The WW3 dataset provides partition parameters for significant wave height, peak period and wave direction, on a

grid with a resolution of ~3 km. Data were available every 3 h, for January 2010–December 2012, although there were some gaps. A total of 7058 directional spectra were modelled using the WW3 data, using the model parameters to construct a Pierson Moskowitz spectrum with $\cos^{2s}$ directional spreading (where $s = 2$).

*2.2. Methodology*

2.2.1. The Seaview Inversion Method

First presented by Wyatt [11], the Seaview inversion (so named because it is being further developed by and is available from Seaview Sensing Ltd.) is an iterative method that begins with an estimated directional spectrum and progressively modifies it to minimise the difference between the calculated and measured Doppler spectra. As explained by Wyatt [20], the initially estimated spectrum was modelled using a Pierson Moskowitz spectrum with a sech directional model, using an empirically-estimated wave height and wind direction derived from the analysis of the first order peaks. The second order radar cross-section expression in Equation (4) was used to calculate the Doppler spectrum using the estimated $S(\vec{k})$, and the output was compared to the measured Doppler spectrum. The spectrum $S(\vec{k})$ was then updated by the rules of the algorithm, and the process was continued until a level of tolerance was achieved between the predicted and measured Doppler spectra. The method was restricted to the frequencies close to the Bragg peaks, where the shorter wave, $\vec{k_2}$, was assumed to be approximately equal to the Bragg wavevector and thus driven by local winds. The initially estimated spectrum could then be used to model $\vec{k_2}$, and then, Equation (4) was simplified and the resulting expression quasi-linear.

At each location, two Doppler spectra, measured by two monostatic radars, were used in order to resolve the directional ambiguity that existed for a single radar measurement. Additionally, for each Doppler spectrum, only values from the sidebands ($\pm 0.6$ Hz) around the dominant Bragg peak were used as input to the inversion method. Furthermore, a signal-to-noise ratio of 15 dB of the largest sideband was a minimum requirement, and each spectrum was normalised by the power in the first order peak, to remove radar effects such as path loss.

2.2.2. The Neural Network Inversion Method

A neural network consists of an *input layer*, a set of *hidden layers* and an *output layer*, where each layer comprises a varying number of *nodes*. The number of nodes in the input and output layers is dependent on the specific problem, and the number in the hidden layers is set by the user. Furthermore, each node is connected to all of the nodes in the previous and following layers by some weight, $w_{kj}^{[l]}$, which signifies the weight between the $j^{th}$ node in the $(l-1)^{th}$ layer and the $k^{th}$ node in the $l^{th}$ layer. At each layer, a bias vector, $\vec{b}^{[l]}$, is also added. Figure 3 depicts a basic neural network.

The goal of a neural network is to learn appropriate values for the weights, connecting the neurons, in order to map accurately some input vector $\vec{x}$ to an output vector $\vec{y}$.

To train the neural network, a *training set* was used, which consisted of input and output data pairs, $(\vec{x}_i, \vec{y}_i)$. The training process included two steps: forward propagation and backward propagation.

In the forward propagation step, an input value $\vec{x}$ (which, for the following expression, is equivalent to $\vec{a}^{[0]}$), was propagated from left to right through the neural network. To do this, at hidden layer $l$, we calculated:

$$\vec{a}^{[l]} = g(\vec{z}^{[l]}),$$

(5)

where:

$$\vec{z}^{[l]} = \vec{w}^{[l]}\vec{a}^{[l-1]} + \vec{b}^{[l]},$$

(6)

for $\vec{z}^{[l]} \in \mathbb{R}^{m \times 1}$, $\vec{w}^{[l]} \in \mathbb{R}^{m \times n}$, $\vec{a}^{[l-1]} \in \mathbb{R}^{n \times 1}$, and $\vec{b}^{[l]} \in \mathbb{R}^{m \times 1}$, where there are $m$ nodes in layer $l$ and $n$ in layer $l - 1$. In Equation (5), $g$ is an *activation function*, which is usually some nonlinear function, which allows the algorithm to learn more complex nonlinear relationships; some common choices include the sigmoid, relu, elu [31] and tanh functions. This process was repeated until the output layer was reached, and then, the generated value, $\hat{y}$, was compared to the training set output value, $\vec{y}$, by some defined metric, here called the *cost function*, which measures the error in the prediction.

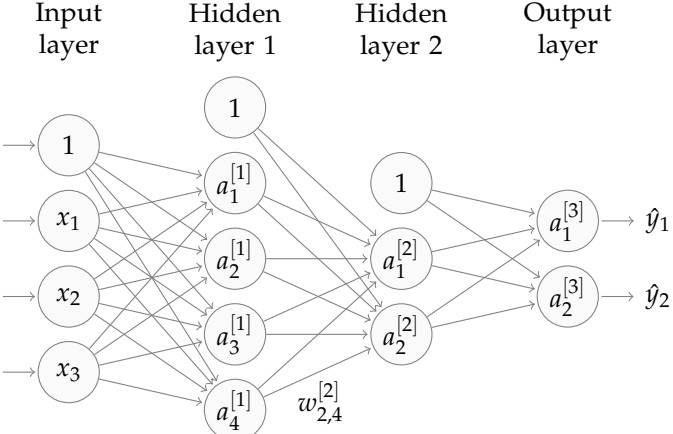

**Figure 3.** Basic neural network architecture with 3 input features, $\vec{x}$, 4 and 2 nodes in two hidden layers and 2 output features, $\hat{y}$. The additional bias nodes are shown at the top of the first three layers. The weights are represented by the lines connecting the nodes, and weight $w_{2,4}^{[2]}$ is shown to highlight the notation.

We then performed backpropagation, where the error was propagated back (i.e., right to left) through the neural network, using the chain rule, to find the derivatives of the cost function with respect to each weight and bias term. The derivatives were used in a minimising algorithm, such as the Adam [32] or RMSProp optimisers, to update the values of the weights such that the prediction error was decreased.

The forward/backward propagation process was repeated until a set number of iterations was executed, or a specific accuracy level was reached. The learning phase was then stopped and the results tested. To ensure the test was unbiased, a *test set* was used, which is a reserved portion of the dataset on which the neural network has not been trained. Using the test set, we can identify if the model is overfitting, which is when the neural network performs significantly better on the training data than the test data, i.e., it fails to generalise. In such a case, we can use *regularization*, which penalises the neural network when it fits too well, to reduce overfitting. The test set is usually smaller than the training set, as the training phase requires as much data, with as much variation, as possible.

The performance of the neural network largely depends on its architecture; therefore, choosing the following parameters is important:

- number of layers;
- number of nodes per layer;
- activation function;
- number of iterations;
- minimising algorithm.

There is no particular rule on how to choose the optimal architecture, it is often a case of investigating the different combinations and finding one that is within a tolerance. A trade-off between time and performance may arise, and the user then has to decide what is more important. A genetic algorithm can be used to search more intelligently through a large set of combinations and is, in fact, used in this work; see the work of Mitchell [33] for an introduction to the subject.

In this work, the goal is to train a neural network (implemented using the TensorFlow framework for Python 3 (https://www.tensorflow.org/)) that can map from radar Doppler spectra to the corresponding ocean spectra. Although it may be possible for a neural network to learn to resolve the directional ambiguity that exists for a single radar, this is yet to be tested. Therefore, to resolve the directional ambiguity, two radars will be used to provide two spectra for each location. Consequently, the neural network will have two radar Doppler spectra as the input and the corresponding directional ocean spectrum as the output.

Depending on where the radars are situated, we define $\phi_1$ and $\phi_2$ as the beam angles, measured clockwise from north, from each radar to a measurement point. As the beam angles are different for each measurement location in the radar coverage area, the resulting Doppler spectra are also different. Therefore, either a different neural network must be trained for every desired location or a single neural network must be able to understand the relationship between $\phi_1$, $\phi_2$ and the related Doppler spectra. In this work, both methods will be tested:

1. the single-location neural network, or *SLNN*, trained in one location;
2. and the multi-location neural network, or *MLNN*, trained to understand the relationship between the radar beam angles $\phi_1$ and $\phi_2$ and the Doppler spectra.

For both neural network experiments, the training/test data were generated in the same way. Beginning with a large dataset of directional ocean spectra, from both the wave buoy and WW3 datasets described in Section 2.1, we used the radar cross-section expression, from Equations (2) and (4), to simulate the corresponding Doppler spectra at a particular location, for both radars. The radar frequencies, the depth of the ocean and the beam angles ($\phi_1$ and $\phi_2$) were fixed, and two 512-point Doppler spectra ranging from $-1.27$ Hz–1.27 Hz were simulated. These simulations should mimic the real data as closely as possible to make the neural network's task less complicated, and therefore, appropriate noise floor and general noise levels were included in the simulations, after analysing the available measured radar data.

Then, any Doppler spectrum input to the neural network (so either the simulated data or the measured validation data) was filtered and processed. An SNR of 10 dB for the second order peaks was set as a minimum requirement for a Doppler spectrum to be used in the neural network, and anything below was discarded, as high noise levels may complicate the neural network model. Any current-induced Doppler shifts were removed (which in this case only applies to the measured radar data) before the resulting Doppler spectra were normalised, so that the first order peaks were set to 1 and $-1$, and the highest Bragg peak was set to a fixed value, here 70 dB. Values $\pm 0.6$ around each peak were used as input to the neural network, which for this experiment gave a 174 point array for each radar, say $\sigma(\eta)_1$ and $\sigma(\eta)_2$ (where $\eta = \omega/\omega_B$). This smaller subset of points was used so that the neural network did not waste resources trying to understand the less important outer parts of the Doppler spectrum, whilst fully enclosing the second order continuum.

The output values, $y_i$, were the directional spectra, and they were interpolated onto a grid of $36 \times 98$, where there were 36 directional values in the range $[0, 2\pi]$ and 98 wavenumber values in the range $[0.004, 0.986]$. The values were then each multiplied by $k$, for scaling purposes, and flattened, to form a vector of a size of 3528, for processing.

In the neural network, an *elu* function, i.e.,

$$g(x) = \begin{cases} x, & x \geq 0 \\ \alpha(\exp(x) - 1), & x < 0 \end{cases},$$

for constant $\alpha$, was added in the final layer to encourage the output spectrum values to be positive. The value of $\alpha$ was set to the TensorFlow-default value of 1, to avoid the cost of tuning another parameter. The cost function was the mean squared error of the predicted and actual values of $S(\vec{k})$; namely:

$$C(\vec{w}, \vec{y}) = \frac{1}{2M} \sum_{i=1}^{M} \left| \hat{\vec{y}}(\vec{w})_i - \vec{y}_i \right|^2,$$ (7)

where $\hat{\vec{y}}_i$ and $\vec{y}_i$ are the *i*th predicted and actual values, respectively, in a total of $M$ values. To find the optimal neural network architecture, a genetic algorithm was used, whose criteria were to minimise the differences in the predicted and actual values in the test set.

Single-Location Neural Network

In training the SLNN, $\phi_1$ and $\phi_2$ were fixed and therefore unnecessary for the training process. Thus, we define $\vec{x}_i$ as the two radar Doppler spectra subsets joined together, $[\sigma(\eta)_1, \sigma(\eta)_2]$, which forms a vector of size 348, and $\vec{y}_i$ as the related directional ocean spectrum $S(\vec{k})$. To validate the method, the location of the simulations was set to the position nearest to the wave buoy where operational radar data were available. At this position, we found that $\phi_1 = 15°$, $\phi_2 = 264°$, and the ocean depth was 53 m; the rest of the radar information used in the simulation process is given in Section 2.1.1.

In this experiment, we used both the WW3-modelled and wave buoy-measured directional spectra in the simulations, as described in Section 2.1. After filtering by the SNR limit, a total of 4804 training examples were available, which were then split into 66%/33% training/test sets for the training phase. After some experimenting, the values shown in Table 2 were decided for the genetic algorithm through which to search.

**Table 2.** The sets of parameters defined for the genetic algorithm to search through, for the SLNN.

| | |
|---|---|
| Number of Nodes | [179, 256, 512, 880] |
| Number of Layers | [6, 7, 8, 9] |
| Minimising Algorithm | [RMSProp, Adam] |
| Activation Function | [elu, relu] |
| Number of Iterations | [2500, 3000, 3500] |

Multi-Location Neural Network

For a neural network being trained to understand the location during the inversion, namely the MLNN, the beam angles $\phi_1$ and $\phi_2$ (scaled to be proportional to the range of the Doppler spectra) must also be passed into the algorithm. Thus, we define $\vec{x}_i$ as the 350-point vector $[\phi_1, \sigma(\eta)_1, \phi_2, \sigma(\eta)_2]$ and, again, $\vec{y}_i$ as the related $S(\vec{k})$.

The test was carried out on a relatively small scale, where 9 locations, including the location of the wave buoy, were chosen, as shown in Figure 2. For the 9 positions, $\phi_1$ ranged between 9° and 42°, and $\phi_2$ ranged between 261° and 288°.

In this experiment, like the SLNN, both the wave buoy-measured and WW3-modelled values of $S(\vec{k})$ were used to simulate the training set. Post filtering, the dataset contained 109,971 data pairs, which were divided into the training/test sets by a 67%/33% split.

Similarly to the SLNN, some experimenting was carried out to decide on the parameters for the genetic algorithm. The resulting parameters are shown in Table 3.

**Table 3.** The sets of parameters for the genetic algorithm to search through for the MLNN.

| | |
|---|---|
| Number of Nodes | [444, 497, 512, 666, 701, 768, 800, 999] |
| Number of Layers | [4, 5, 6, 7, 8, 9] |
| Minimising Algorithm | [RMSProp, Adam] |
| Activation Function | [elu, relu] |
| Number of Iterations | [2000, 2500, 3000, 3500] |

## 3. Results

To measure the accuracy of the inversion, the values of mean wave direction, $\theta_m$, peak wave direction, $\theta_p$, peak period, $t_p$, energy period, $t_E$, and significant wave height, $h_s$, defined in Appendix A, were calculated from the neural network algorithm-predicted directional wave spectra and compared with those measured by the wave buoy and the Seaview method. To compare the results quantitatively, the root mean squared error (RMSE) and correlation coefficient (CC) of each parameter were calculated and presented for both the neural network and Seaview inversion methods. Scatter and time series plots are also given, to present the results qualitatively.

For both the SLNN and MLNN, each genetic algorithm was run for a total of 96 h, on a GPU in a high performance computing (HPC) cluster at the University of Sheffield, in its search for the optimal neural network parameters. In training the SLNN, the algorithm tested 173 combinations before the parameters were chosen, as summarised in Table 4. In the case of the MLNN, 67 combinations were tested before the parameters (also shown in Table 4) were chosen, noting that a lower number was tested within the time frame due to the larger training data size and the larger neural network required.

**Table 4.** The hyperparameters chosen by genetic algorithms for the single-location neural network (SLNN) and multi-location neural network (MLNN) inversion methods.

|  | SLNN | MLNN |
| --- | --- | --- |
| Number of Layers | 8 | 9 |
| Number of Nodes | 880 | 999 |
| Minimising Algorithm | Adam | Adam |
| Activation Function | elu | relu |
| Number of Iterations | 3000 | 3000 |

### 3.1. SLNN Results

The numerical results for the SLNN training and test sets are given in Table 5, and scatter plots are shown in Figure 4. The results showed a strong correlation between the simulated and predicted values of each ocean parameter, showing that SLNN can accurately invert simulated HF radar data.

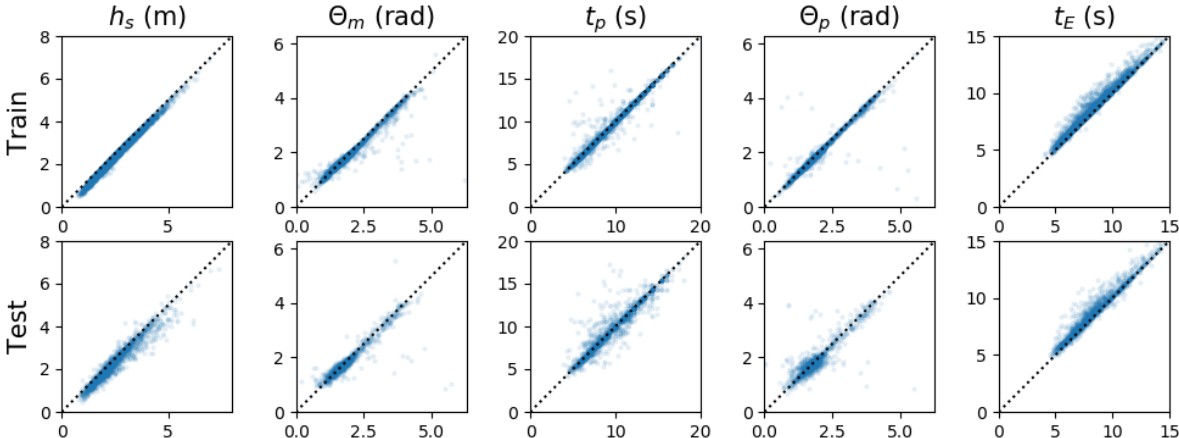

**Figure 4.** Scatter plots showing the results of the SLNN on the training (**top**) and test (**bottom**) datasets. The simulated values are shown on the *x* axes and the predicted values on the *y* axes.

The results of the trained neural network, when applied to the Wave Hub HF radar data, are shown in Table 6. The monthly comparisons of the ocean spectrum parameters are shown in Figures 5 and 6. Note that for this inversion method, there were 644 Doppler spectra available for inversion; this is more than the Seaview method due to a difference in the SNR restriction levels, however, a comparison of the same dataset is given in Table 6.

The results are similar to those of the test set, where both directional parameters, $\theta_m$ and $\theta_p$, were predicted accurately, with small error and strong correlation. In comparison to the Seaview inversion, the inferred directional values appeared to be more accurate, with fewer outliers. On the other hand, the value of $t_E$ was generally overestimated, an attribute not shared with the Seaview inversion, signalling that the neural network inversion was less accurate at predicting the power in longer ocean waves. The derived values of $h_s$ were similar to the wave buoy and Seaview values at lower wave heights, but were underestimated at the higher wave heights, approximately when $h_s > 4$; a discussion about this difference is presented in Section 4. The comparison of $t_p$ shows that the Seaview and SLNN methods performed similarly, where the SLNN results showed more variance, but fewer significant outliers.

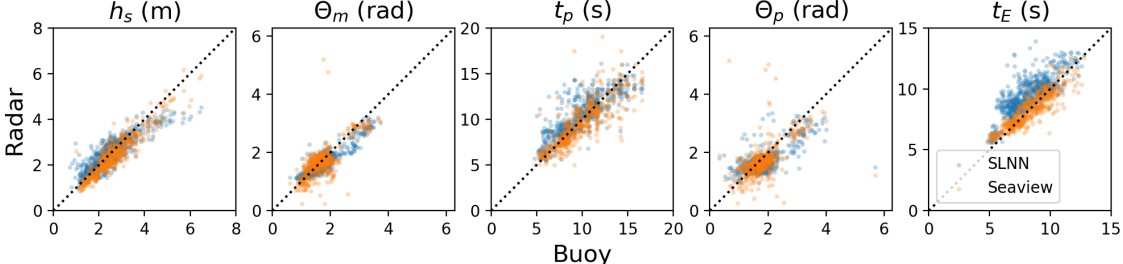

**Figure 5.** Scatter plots showing the results of the SLNN inversion on the Wave Hub HF radar data in November 2012. The wave buoy data, for the same period, are shown on the *x* axis of each plot, and the inverted radar data are on the *y* axis.

**Table 5.** Numerical results of the SLNN and MLNN inversion methods, on the training and test datasets. The correlation coefficients (CC) and root mean squared errors (RMSE) are given. The mean and standard deviation of each parameter, for both the training and test datasets, are also given to provide further context for the RMSE values.

| | SLNN | | | | MLNN | | | | Data Statistics | | | |
|---|---|---|---|---|---|---|---|---|---|---|---|---|
| | **Train** | | **Test** | | **Train** | | **Test** | | **Train** | | **Test** | |
| | **CC** | **RMSE** | **CC** | **RMSE** | **CC** | **RMSE** | **CC** | **RMSE** | **Mean** | **SD** | **Mean** | **SD** |
| $h_s$ (m) | 1.0 | 0.18 | 0.96 | 0.31 | 0.9 | 0.44 | 0.9 | 0.44 | 2.47 | 0.96 | 2.45 | 0.96 |
| $\theta_m$ (rad) | 0.96 | 0.15 | 0.91 | 0.19 | 0.86 | 0.28 | 0.87 | 0.28 | 1.89 | 0.65 | 1.85 | 0.61 |
| $t_p$ (s) | 0.98 | 0.52 | 0.94 | 1.03 | 0.87 | 1.69 | 0.86 | 1.69 | 9.36 | 2.81 | 9.60 | 2.89 |
| $\theta_p$ (rad) | 0.95 | 0.16 | 0.88 | 0.26 | 0.82 | 0.32 | 0.82 | 0.33 | 1.90 | 0.64 | 1.85 | 0.61 |
| $t_E$ (s) | 0.99 | 0.54 | 0.97 | 0.63 | 0.91 | 1.12 | 0.91 | 1.12 | 08.14 | 2.07 | 8.28 | 2.12 |

**Table 6.** Numerical results of the Seaview, SLNN and MLNN inversion methods, tested on the Wave Hub HF radar data. The Seaview results are for the inversion of 501 Doppler spectra, whereas the SLNN and MLNN results are for the inversion of 644 Doppler spectra. The correlation coefficients (CC) and root mean square errors (RMSE) are given. The mean and standard deviation of each parameter, for the wave buoy dataset, are also given to provide further context for the RMSE values.

| | Seaview | | SLNN | | MLNN | | Wave Buoy Statistics | |
|---|---|---|---|---|---|---|---|---|
| | **CC** | **RMSE** | **CC** | **RMSE** | **CC** | **RMSE** | **Mean** | **SD** |
| $h_s$ (s) | 0.93 | 0.4 | 0.87 | 0.47 | 0.8 | 1.51 | 2.50 | 0.94 |
| $\theta_m$ (rad) | 0.75 | 0.36 | 0.91 | 0.26 | 0.83 | 0.37 | 1.78 | 0.60 |
| $t_p$ (s) | 0.83 | 1.47 | 0.82 | 1.56 | 0.75 | 3.18 | 9.77 | 2.55 |
| $\theta_p$ (rad) | 0.59 | 0.45 | 0.70 | 0.42 | 0.62 | 0.47 | 1.85 | 0.59 |
| $t_E$ (s) | 0.85 | 0.85 | 0.80 | 1.64 | 0.82 | 2.53 | 8.18 | 1.62 |

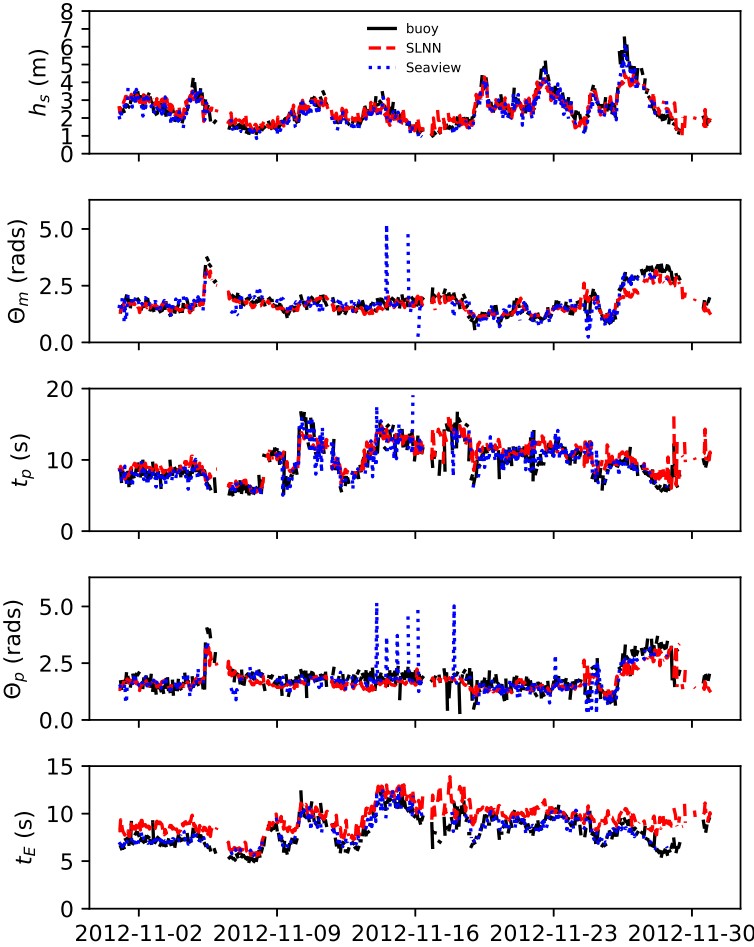

**Figure 6.** Time series plots, for November 2012, showing the results of the Seaview inversion on the Wave Hub HF radar data against the values measured by the wave buoy.

## 3.2. MLNN Results

The results of the inversion, when applied to the training and test sets, are shown numerically in Table 5 and visually in Figure 7. The results were similar for both the training and test sets, but showed that the neural network was not performing particularly well on either dataset.

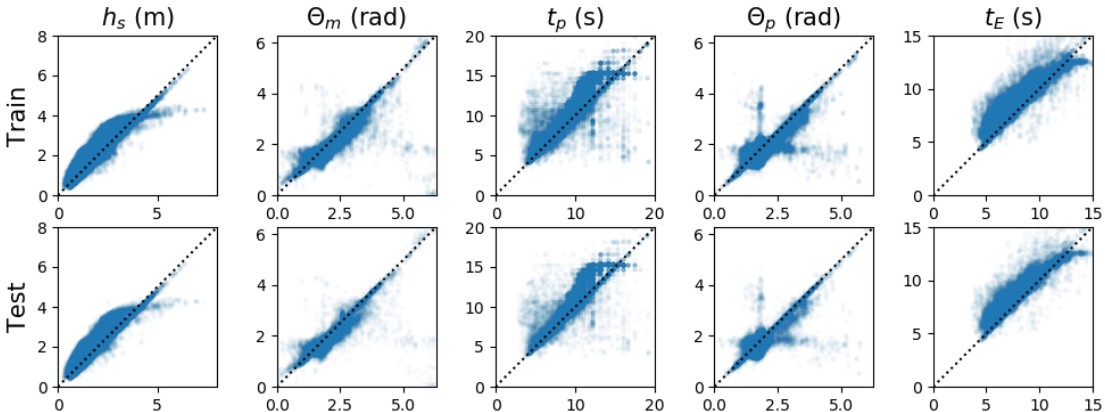

**Figure 7.** Scatter plots showing the results of the MLNN on the training (**top**) and test (**bottom**) datasets. The simulated values are shown on the *x* axis and the predicted values on the *y* axis.

The results of the trained neural network, when applied to the Wave Hub HF radar data, are shown in Table 6 and Figures 8 and 9. These results showed levels of under-performance similar to the training/test sets, with only $\theta_m$ inferred reasonably well. Possible reasons and remedies for this are discussed in Section 4, and solving this issue was imperative to this inversion method being a viable inversion option.

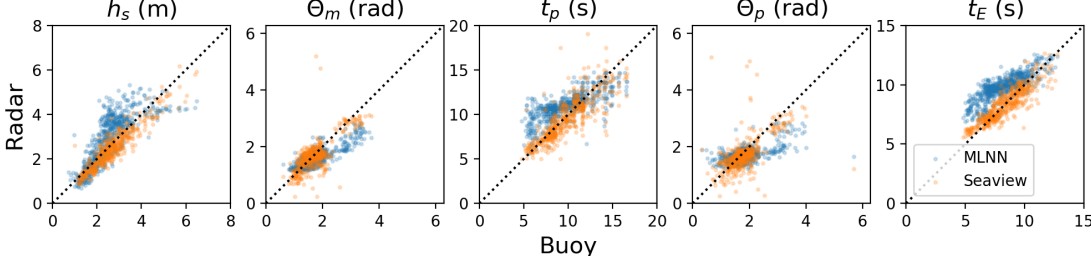

**Figure 8.** Scatter plots showing the results of the MLNN inversion on the Wave Hub HF radar data in November 2012. The wave buoy data, for the same period, are shown on the *x* axis of each plot, and the inverted radar data are on the *y* axis.

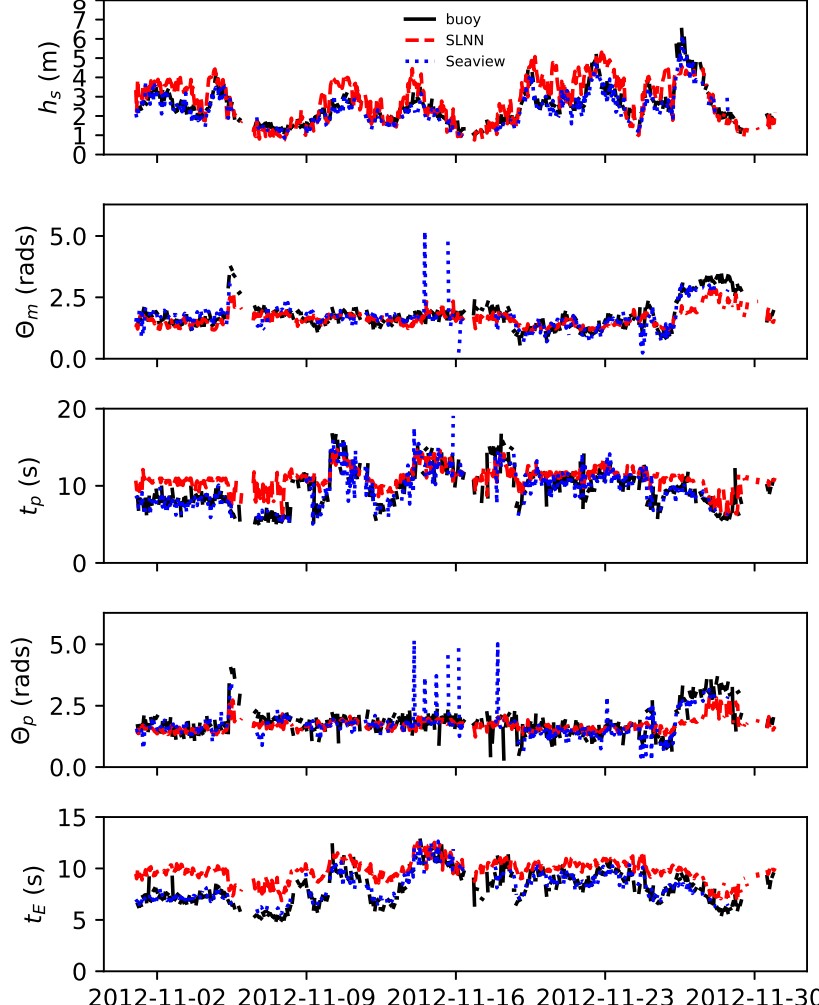

**Figure 9.** Time series plots, for November 2012, showing the results of the MLNN inversion on the Wave Hub HF radar data. The wave buoy data and Seaview inversion results are shown, for comparison.

An example of derived wave heights and mean wave direction for the nine trained locations is shown in Figure 10, and an inversion of Doppler spectra from a larger expanse of the ocean, for the same time, is shown in Figure 11. A Seaview inversion, for the same time and date of these examples, is shown in Figure 12. The Seaview map shows more directional variation than the MLNN map, which seems inclined to predict the mean value of $\theta_m$. A discussion of possible ways to rectify this issue is given in Section 4, although the MLNN-derived values of $h_s$ are encouraging.

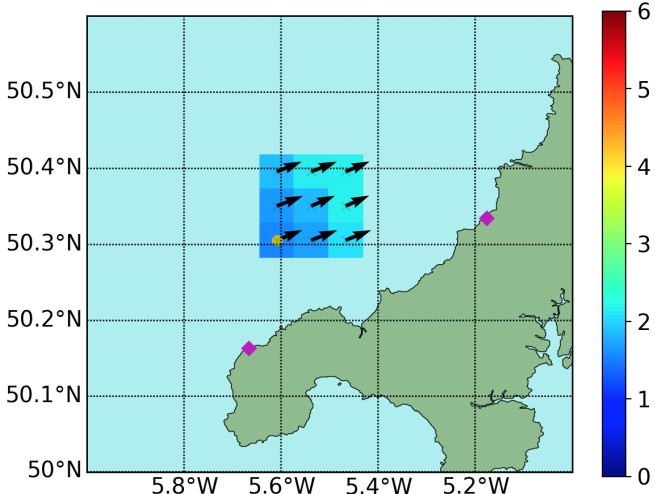

**Figure 10.** An example of MLNN-derived values at each of the nine locations that the MLNN was trained at, on 20 November 2012 at 03:00. Significant wave height, measured in metres, is shown by the background colour of each cell, and the mean wave direction is shown by the black arrows.

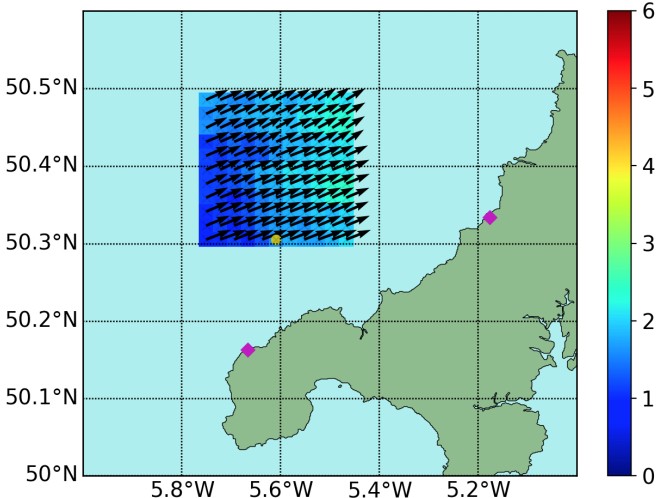

**Figure 11.** An example of a whole field inversion, for 20 November 2012 at 03:00, using the MLNN inversion method, where Doppler spectra from untrained locations were also inverted. Significant wave height, measured in metres, is shown by the background colour of each cell, and the mean wave direction is shown by the black arrows.

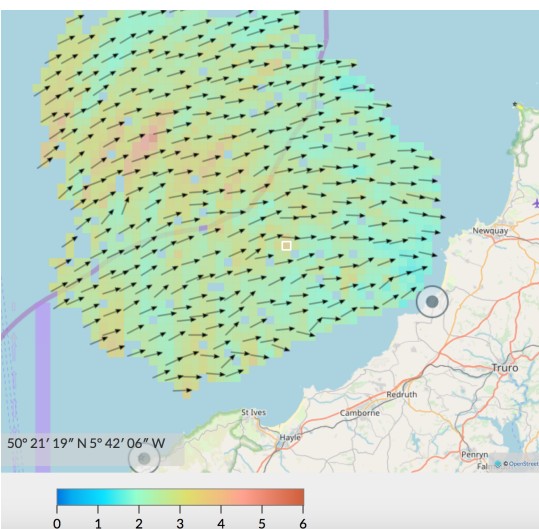

**Figure 12.** An example of the whole field inversion, for 20 November 2012 at 03:00, using the Seaview inversion method on the Wave Hub HF radar data (taken from https://phillips.shef.ac.uk/pub/svdv/whc/). Significant wave height, measured in metres, is shown by the background colour of each cell, and the mean wave direction is shown by the black arrows.

## 4. Discussion

The SLNN inversion showed generally good agreement with the wave buoy and the Seaview inversion method and showed the potential of machine learning algorithms in the field of remote sensing using HF radar. The derived values of $\theta_m$ were particularly encouraging; however, the value of $t_E$ seemed to be overestimated. At sea states above approximately 4 m, the neural network was underestimating $h_s$, which was perhaps due to a limitation in the perturbation analysis used in the derivation of the radar cross-section. In the derivation, small slopes in comparison to the radar wavelength were assumed. Therefore, for high seas with large significant wave height, the theory broke down, and the simulations that the neural network was trained on did not represent the real data well. Wyatt et al. [34] explained how, in the Seaview method, the first order Bragg peaks were scaled, such that the second order continuum was increased in comparison to the first, for a given wave height. The scaling was based on the work of Creamer et al. [35], who showed that a second order contribution (due to the combination of first and third order ocean waves) was missed in the derivation of the radar cross-section by Barrick [2,3] and was significant at larger wave heights. The improvement in the results when the scaling was included was shown in the work of Wyatt et al. [34] and is something that could be included in future work, by modifying the simulations appropriately, to improve the accuracy in predicting $h_s$.

Interestingly, the numerical comparison of the SLNN and Seaview inversions for the same dates (shown in Table 7) did not affect the SLNN results much, indicating that the neural network inversion was performing well on the data with a lower SNR.

The MLNN inversion, although providing encouraging results, did not perform as well as the SLNN inversion. From looking at the results of the training and test datasets, the introduction of the radar beam angles clearly affected the model accuracy and hence why the results were not as good as the SLNN. In Figure 11, the MLNN-derived values of significant wave height and mean wave direction are shown at locations the neural network was not trained. The values of $\theta_m$ were all very similar, in contrast to the more varied values derived by the Seaview inversion, shown in Figure 12. A possible reason for this was that when the neural network was trained, the algorithm got stuck in a local minimum and learned to predict the average values for $\theta_m$ and $\theta_p$. This was further suggested by the results of the inversion of the entire dataset, shown in Figure 8, where $\theta_p$ in particular seemed inclined to predict the average value of the dataset.

**Table 7.** Numerical results of a direct comparison of 501 inversions, of the Seaview, SLNN and MLNN inversion methods, tested on the Wave Hub HF radar data. The correlation coefficients (CC) and root mean squared errors (RMSE) are given. The results are for a subset of the Wave Hub data, on which the Seaview inversion results are based. The mean and standard deviation of each parameter, for the wave buoy dataset, are also given to provide further context for the RMSE values.

| | Seaview | | SLNN | | MLNN | | Wave Buoy Statistics | |
|---|---|---|---|---|---|---|---|---|
| | **CC** | **RMSE** | **CC** | **RMSE** | **CC** | **RMSE** | **Mean** | **SD** |
| $h_s$ (s) | 0.93 | 0.4 | 0.88 | 0.46 | 0.77 | 1.49 | 2.66 | 0.93 |
| $\theta_m$ (rad) | 0.75 | 0.36 | 0.88 | 0.27 | 0.68 | 0.4 | 1.71 | 0.52 |
| $t_p$ (s) | z 0.83 | 1.47 | 0.84 | 1.46 | 0.77 | 3.15 | 9.82 | 2.52 |
| $\theta_p$ (rad) | 0.59 | 0.45 | 0.69 | 0.4 | 0.7 | 0.43 | 1.80 | 0.56 |
| $t_E$ (s) | 0.85 | 0.85 | 0.83 | 1.48 | 0.83 | 2.52 | 8.26 | 1.61 |

Therefore, more work must be done on successfully including the radar beam angle in the neural network algorithm. The main advantage of HF radar is its ability to provide measurements across a larger expanse of the ocean, as opposed to the fixed location of a wave buoy. As the neural network can work in a single location, there is no reason why it should not also be able to learn the relationship between $\phi_1$ and $\phi_2$ and the Doppler spectra to the same level of accuracy. This is seemingly a case of getting the correct neural network architecture and having a big enough dataset for the training process. A training dataset with the locations spaced further apart could be used, and perhaps a larger neural network would also be beneficial; additionally, a convolutional neural network may be more appropriate for this task.

There were a number of variables not yet experimented within the training phase, which may impact the accuracy of the neural network. The SNR minimum was fixed at 10 dB in this experiment; however, in the Seaview inversion, for example, the SNR minimum was 15 dB. Similarly, the points of the Doppler spectrum of 0.6 on either side of each Bragg peak was input into the neural network. Perhaps valuable information was being missed in the training phase or useless information was being passed in and wasting resources. The issue with neural networks is their "black box" nature, whereby it can be difficult to interpret the results. There are a vast number of other machine learning methods that could be used for this task, such as a random forest regression model, which may be more interpretable, and a comparison study, discovering which method is most appropriate, would be interesting.

The expense of the inversion process in both time and computing resources was fairly high. Other methods, for instance the Seaview inversion method, require no prior knowledge of the radar data. One just has to set variables, such as radar frequency and location, and then, the algorithm is ready to be implemented. As the neural network procedure is dependent on factors such as the locations of the radars, new simulations and hence another training process must be carried out for each new radar site. If a radar were to be moved, the whole process must be carried out again, meaning a few days of operational downtime would occur. Training a neural network to recognise all different radar frequencies and angles would be theoretically possible, although an exceptional amount of computing power and a much larger neural network and dataset would be necessary.

The neural network approach has proven itself to be capable of inverting HF radar Doppler spectra, in a single location, when the radar quality is good. The next step is to test the method in a more challenging situation, where the existing methods perform poorly, for instance for a bistatic phased array radar with low spatial resolution. Grosdidier et al. [22] showed that for their particular bistatic radar, the data were not accurately modelled by the radar cross-section, due to the limitations of the radar configuration at their site. A simulation method that includes these radar effects would better represent this type of data. Then, training a neural network on a large dataset of Doppler spectra, simulated in this way, should theoretically enable us to invert these more complicated Doppler spectra.

## 5. Conclusions

In this work, we showed that a neural network can accurately invert HF radar Doppler spectra to obtain directional wave spectra. Key ocean statistics, namely mean wave direction, peak wave direction, energy period, peak wave period and significant wave height, were all obtained from measured radar data and the results compared with the values obtained by the Seaview inversion method and those measured by a wave buoy.

However, a number of issues with this new approach were identified: (1) the radar beam angle was not successfully included in the neural network inversion method, meaning that Doppler spectra measured over a large area of the ocean cannot be accurately inverted; and (2) the expense of training the algorithm was high in both resources and time. Nonetheless, the method can potentially be used more easily with radar data affected by sidelobes or poor spatial resolution since these effects can be included in the neural network training process.

**Author Contributions:** Conceptualization, R.L.H. and L.R.W.; methodology, R.L.H. and L.R.W.; software, R.L.H. and L.R.W.; validation, R.L.H. and L.R.W.; formal analysis, R.L.H. and L.R.W.; investigation, R.L.H. and L.R.W.; resources, R.L.H. and L.R.W.; data curation, R.L.H. and L.R.W.; writing, original draft preparation, R.L.H.; writing, review and editing, L.R.W; visualization, R.L.H. and L.R.W.; supervision, L.R.W.; project administration, L.R.W.; funding acquisition, R.L.H. and L.R.W.

**Funding:** This research was funded by the Engineering and Physical Sciences Research Council (Grant Number EP/M506618/1). The APC for this work has been waived.

**Acknowledgments:** We are grateful to Daniel Conley and Guiomar Lopez at the University of Plymouth for the Wave Hub HF radar and wave buoy datasets.

**Conflicts of Interest:** R.L.H. declares no conflict of interest. L.R.W. is the Technical Director of Seaview Sensing Ltd, but her role in this paper is as a University of Sheffield Professor, and as such, she has only made scientific inputs and judgements on the work. The funders had no role in the design of the study; in the collection, analyses, or interpretation of data; in the writing of the manuscript; nor in the decision to publish the results.

## Appendix A. Ocean Spectrum Statistics

Firstly, the non-directional spectrum, $E(f)$, is calculated from $S(f, \theta)$ by:

$$E(f) = \int_0^{2\pi} S(f, \theta)\, d\theta \tag{A1}$$

and the first Fourier coefficients $a_1$ and $b_1$ (see Kuik et al. [36]) are:

$$a_1 = \int_0^{2\pi} S(f, \theta) \cos\theta\, d\theta \qquad\qquad b_1 = \int_0^{2\pi} S(f, \theta) \sin\theta\, d\theta. \tag{A2}$$

From Tucker [37], the significant wave height, $h_s$, is defined as:

$$h_s = 4\sqrt{\int E(f)\, df}. \tag{A3}$$

To measure where the majority of the energy is in the spectrum, peak period, $t_p$, which is the inverse of the spectral peak frequency, $f_p$, and the energy period, $t_E$, which measures roughly where the power is based in frequency, as defined by Wyatt [20], are used. They are defined as:

$$t_p = \frac{\int_{f_1}^{f_2} E(f)\, df}{\int_{f_1}^{f_2} f E(f)\, df}, \tag{A4}$$

where $E(f_1) = E(f_2) = 0.8 E_{max}$ and:

$$t_E = \frac{\int f^{-1} E(f)\, df}{\int E(f)\, df}. \tag{A5}$$

To measure directional information, the peak direction, $\theta_p$, which is the direction of the most energetic waves in the spectrum, is defined as:

$$\theta_p = \arctan\left(\frac{b_1(f_p)}{a_1(f_p)}\right),\tag{A6}$$

for $f_p = 1/t_p$ and $a_1$ and $b_1$ as in Equation (A2). The mean wave direction is defined as:

$$\theta_m = \arctan\left(\frac{\int b_1(f)\,df}{\int a_1(f)\,df}\right).\tag{A7}$$

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
