# Peer review of "Inversion of HF Radar Doppler Spectra Using a Neural Network"

_jmse, doi:10.3390/jmse7080255_

Round 1

Reviewer 1 Report

In the paper entitled: “Inversion of HF Radar Doppler Spectra using Neural Network” the authors describe a method, based on neural network, to obtain the directional ocean spectrum from HF radar Doppler Spectra.

The arguments are well introduced, and the method and results described clearly. The results point out how the method can be efficient in inverting HF Radar Doppler spectra and allowing to get the direction wave spectra. The authors also show how neural network can work even when the data spatial resolution is poor or large sidelobes occur. 

I found the work well structured and although improvements can be almost always made, I think that the paper can be accepted in the present form.  

Author Response

Thank you for the review. Kind regards, Rachael

Reviewer 2 Report

OVERALL
The paper is a worthy contribution and should be published. Most of my comments are to clarify, simplify and perhaps improve the presentation. Some work needs to be done improve the readability and content of the results section, and to reduce redundancies in the figures, perhaps making both more concise.  

GENERAL COMMENTS
- how little training data can you get away with? Could you simplify the problem by using WW3 and HFR to train and then validate at a buoy? This might make the method more generally applicable
- You might make it a bit more clear how close to the seaview results the SLNN results are .. it appears that you have nearly accomplished in a few years with the use of NN, what has taken  decades with other methods
- its not clear why you needed to simulate the HF radar data, or how this was done. References for computing the RCS are given but no refs for simulating spectra
- You may wish to include more specific results in the abstract, such as how this method compares with previous and a summary of the main results (that is, how well you get waves from HF using neural network methods).

SPECIFIC COMMENTS

Eqn 2 define delta

check eqn 3 definition (full wave eqn?)

what about the work of the canadians? (is Walsh ref one of them?)

Fig 1: Why the big difference near the nulls? Any references for the simulation (eg there is
a citation for the RCS, but this is just a part of the simulation?

Line 53: "Large sidelobes" it has not been explicitly meantioned that this is a PA using BF?

Line 57: suggest rephrasing that the network "should be able to understand"

Line 71: you use a model and a buoy ...

Table 1: suggest using degrees instead of radians for angular units

line 104: suggest rewording: "WAVEWATCH III (WW3) data"

line 126: Not clear what is meant by this sentence. Also, direction ambiguity has not been explained

line 140, eqns 5,6: suggest not renaming the variable, makes a difficult topic more confusing. Also, it would be helpful to give examples of the dimensions of these variables (which I assume are vectors)

line 156: You may want to include mention of the relative sizes of the test and training data sets here, and a discussion of the
possibility of 'overfitting' if it's appropriate for neural networks

line 165: the use of genetic algorithm is presented here as a hypothetical, where later on it is revealed that this is what was used

line 182: This is a bit confusing because it says the data was collected but then discusses how the data was simulated

line 192: Suggest avoiding the anthropomorphism of the algorithm. High noise levels probably cause it to attribute structure (or variance?) to the noise? Also, it is not clear why, or how you remove the current shifts and why you do this. Is this part of the simulation?

line 206: Suggest moving the use of tensorflow up toward the beginning of section 2.2.2. You might put this in the abstract also. It seems like a term that people would use in a search?

line 216: Suggest moving the description of the training/test data set sizes up toward the begining of this section

Figs 4 and 5: you may want to drop these figures, since they do not seem to add much over what is shown in Fig. 3, and the lack of bias terms in them suggests they are incomplete anyway.

line 221: not clear what is meant by 'in line with the Doppler spectra'. I think I know what you mean, but you may want to say it differently.

Fig 6: suggest combining this with Fig. 2

Table 3: I don't think RMSProp and Adam have been defined

line 237:  the words "to 2 decimal places" seem unnecessary (elsewhere as well)

Results section: Rather they stating where the results can be found, I would suggest summarizing the main results, and identifying some specific examples. Suggest explaining to the reader what was found rather than having the reader figure it out for themselves from the tables and figures - draw the new knowledge out from the resulting data.

Section 3.1: Perhaps you do not need to discuss the Seaview results here in this way?

Line 247: More information about the computer used would be helpful for readers trying to assess the computational requirements. Is the GPU in a desktop or a big server somewhere?

Figs 7 and 12: these show a lot of redundant content and it seems they could be combined in to one large figure showing the lines with 3 colors. Suggest reducing the axis scales also to better show the results (and their differences).

Fig 9: Suggest showing this along side a figure like Fig. 17, but for the SLNN data (why is there no map of SLNN data?) You may want to de-emphasize the MLNN results by cutting Figs. 16 and 17.

Table 4: Suggest moving this to the methods section?

Table 5: Suggest breaking this into two tables, one for correlation coefficients and one for RMSE. It would be simpler also to have the training results in their own table. The units need to be shown somewhere also - there is a big difference between an RMSE of 0.19 radians vs degrees.

Table 6: Same suggestions as for Table 5, but in addition, you may want a separate table for the bracketed results - these are a fundamental test of the method since it's essentially showing how well it observes real waves, I wouldn't bury the results in part of a table. You may also want to put some of these in the abstract or summarize them there.

Table 6 and Fig 12: Do the statistics of the Seaview data include the obvious outliers for the theta measurements? These would bias the RMSE values

Seems that Figs 8, 10 and 11 can be combined, with a clear description of how they differ. Similarly for Figs 13 and 14

Fig 15: Combine these as was suggested for Figs 12

Lines 272-286: you may wish to add a speculation about what a better training data set would remedy the limitations you describe. Can you just use HF data and WW3 without simulation HF data in the middle?

Line 306: suggest rewording "variables not yet varied"

Line 312: Suggest speculating on the use of different machine learning methods (e.g. support vector machines) if you have any previous experience with these and whether they may provide more insight if they are less of a black box. Also, are you suggesting that an improved understanding of the physics is needed? If so, please state this since that is a useful conclusion.

Line 324: Suggest rewording the sentence beginning with "However" - what does "surplus to requirements" mean?

Author Response

Thanks for your thorough review - very good comments and I think they've greatly improved the work. 

•General comments:

- how little training data can you get away with? Could you simplify the problem by using WW3 and HFR to train and then validate at a buoy? This might make the method more generally applicable- You could definitely just train using the ww3 and simulated radar data; I began with that and got OK results. However, the spectra are a bit too 'perfect' because they're modelled and really you should add noise is a realistic way - I was running low on PhD time so I didn't want to get into simulating realistic noise and as the buoy data was available, I just used that.

- You might make it a bit more clear how close to the seaview results the SLNN results are .. it appears that you have nearly accomplished in a few years with the use of NN, what has taken  decades with other methods

I've attempted to strengthen the results comments a little bit, also added to the abstract.

- its not clear why you needed to simulate the HF radar data, or how this was done. References for computing the RCS are given but no refs for simulating spectra 

I've added in a little bit about the simulation code... it's not open source due to possible uni restrictions but, hopefully, it will be one day. I simulated the data to avoid needing a wave buoy for a long period of time in the radar coverage area - it was just to show it is possible - mentioned in the intro about it being a standalone method and not reliant on other devices like the other NN methods.

- You may wish to include more specific results in the abstract, such as how this method compares with previous and a summary of the main results (that is, how well you get waves from HF using neural network methods). 

I've added a bit into the abstract with a specific result - hopefully it's ok within the 200 word limit.

•Specific comments (I've amended the work appropriately if a response is not in the list below):

-check eqn 3 definition (full wave eqn?)

 it's correct for the specific wavenumber that it's referring to

-what about the work of the canadians? (is Walsh ref one of them?)

Yes, Walsh is one of them.

Table 4: Suggest moving this to the methods section?

I didn't move it as the parameters are technically part of the results - they were found by running the algorithm.

-Fig 9: Suggest showing this along side a figure like Fig. 17, but for the SLNN data (why is there no map of SLNN data?) You may want to de-emphasize the MLNN results by cutting Figs. 16 and 17.

The SLNN is trained in one location and therefore it would be pointless trying to invert Doppler spectra, which would be different, in the other locations. The MLNN was trained for the multiple locations and so I wanted to show what the inversions looked like and that there's still a lot of work to be done!

-Table 5: Suggest breaking this into two tables, one for correlation coefficients and one for RMSE. It would be simpler also to have the training results in their own table. The units need to be shown somewhere also - there is a big difference between an RMSE of 0.19 radians vs degrees.

I didn't separate the RMSE/CC for presentation reasons - but if you feel strongly about it I will do it, the results just seem quite spread out already

Thanks,

Rachael

Reviewer 3 Report

The authors estimated wave parameters using the Neural network (N-N).
The advantage of the method is that wave parameters can be estimated
even when the relationship between the wave spectrum and Doppler spectrum
is unclear, for example in the case of high wave heights.
The N-N is promising for developing wave estimation from HF radar,
This study doesn't take advantage of the N-N, although it is noted
in Line  59-61.
If the objective of the paper is the inversion of equation (2)
using  the N-N, the validation data should be simulated wave data.
Also, the validation of test data in training area (Fig 16)
and outside of training area (in Fig 17 but out of Fig 16)
should be done separately.

Other comments
(1) Fig 2: Please indicate water depth.
(2) Section 2.1.2, Table 1: What is the simulated wave spectra ?
Are they output of WW3 at the buoy location ?
(3) Section 2.1.3: Please describe about model output.
What is the model domain ?
What wind data is used ?
How is the accuracy of WW3 output wave parameters ?
(4) Line 249, Line 260 (table 4):
The parameters of the N-N are selected, probably because
they showed the best performance.
Are the accuracy for the training data related with
the accuracy for the test data ?
For example, if a group of parameters
showed the best performance in the training data,
does it also show the best performance in the test data ?
(5) Table 5,6: Please indicate number of data for comparison.
(6) Most of the wave spectra as training data set are simple
parametric form (Line 110).
As a result, are the wave spectra by the N-N also simple form ?

Author Response

The N-N is promising for developing wave estimation from HF radar,
This study doesn't take advantage of the N-N, although it is noted 
in Line  59-61. 

Sorry, I do not understand this comment. We did use a neural network to invert measured radar Doppler spectra and presented the results.

If the objective of the paper is the inversion of equation (2) 
using  the N-N, the validation data should be simulated wave data.

The real test of this method is to invert measured radar data, as the goal would be to use this method operationally. Therefore, it seemed more insightful to present the results of inverted measured radar data, as opposed to more simulated data. 

Also, the validation of test data in training area (Fig 16)
and outside of training area (in Fig 17 but out of Fig 16)
should be done separately.

I've put some more words around the introduction of these plots which will hopefully help.

•Other comments
(1) Fig 2: Please indicate water depth.

Unfortunately, we do not have accurate depth data for the overall location - we used the depth given by the Wave Hub group.

(2) Section 2.1.2, Table 1: What is the simulated wave spectra ? Are they output of WW3 at the buoy location ?

The simulated wave spectra are the buoy data used for simulating the radar data, used to train the neural network. We got two different datasets, one for validation and one for simulation - to avoid bias.

(3) Section 2.1.3: Please describe about model output.
What is the model domain ? 
What wind data is used ?
How is the accuracy of WW3 output wave parameters ?

We didn't use wind data, we used significant waveheight and peak period in the Pierson Moskowitz spectrum model. The accuracy is of the WW3 data doesn't matter too much, as it's only for training the neural network. In general, the directions are similar but once in the Pierson Moskowitz model, it's inevitably different.

(4) Line 249, Line 260 (table 4):
The parameters of the N-N are selected, probably because they showed the best performance.
Are the accuracy for the training data related with the accuracy for the test data ?
For example, if a group of parameters showed the best performance in the training data,  does it also show the best performance in the test data ? 

Yes, the NN parameters are chosen based on which perform best on the test set. The ones that perform best on the training set overfit; i.e. they do not generalise well to unseen data, such as the test set. This is why I chose the parameters based on the test set.

(5) Table 5,6: Please indicate number of data for comparison.

Added in - thanks.

(6) Most of the wave spectra as training data set are simple 
parametric form (Line 110). 
As a result, are the wave spectra by the N-N also simple form ?

No, because a wave buoy data set was also used, which was more 'realistic'.

Round 2

Reviewer 3 Report

(1) The goal of this paper is to estimate wave spectra (parameters)
operationally, rather than the inversion of equation (4).
On the other hand, the trained data are not measured Doppler spectra
but simulated Doppler spectra except on the buoy point,
therefore, the accuracy of WWW spectra does not matter.
It seems to be inconsistent, and  accurate WWW spectra
and measured Doppler spectra would be used for the goal.
This reason should be described or discussed.
(2) Equation after Line 216: How the value of alpha is determined?
(3) Fig.7: Why are there bias at the higher waveheights in the MLMM,
although most of the training data are simulated data,
and the bias is not clear in the SLNN (Fig.4) ?
This reason should be discussed.

Author Response

Thanks for your comments.

(1) The goal of this paper is to estimate wave spectra (parameters)
operationally, rather than the inversion of equation (4).
On the other hand, the trained data are not measured Doppler spectra
but simulated Doppler spectra except on the buoy point,
therefore, the accuracy of WWW spectra does not matter.
It seems to be inconsistent, and  accurate WWW spectra 
and measured Doppler spectra would be used for the goal.
This reason should be described or discussed.

The trained data at the buoy point are still simulated Doppler spectra. The only time that we use measured Doppler spectra, is in the validation of the method when there is wave buoy data to compare to the directional spectrum predicted by the neural network. I didn't want to train the NN on real data as then the method relies on the wave buoy as discussed in the intro, which is not ideal. Therefore, because it is just training data, the more variation, the better, and that's why the accuracy of the WW3 data doesn't matter; it still simulates Doppler spectra, even if it is not accurate to a specific time and day.

To be clear:

training set: ww3/ wave buoy spectra used to simulate Doppler spectra

test set: ww3/ wave buoy spectra used to simulate Doppler spectra

validation: radar measured Doppler spectra and the directional spectra, measured by the wave buoy. 

(2) Equation after Line 216: How the value of alpha is determined?

Added in (kept as the default value of 1 to avoid having too many parameters to tune).

(3) Fig.7: Why are there bias at the higher waveheights in the MLMM,
although most of the training data are simulated data,
and the bias is not clear in the SLNN (Fig.4)?
This reason should be discussed.

I've added a comment into the discussion - thanks.